# Current Knowledge on the Healing of the Extraction Socket: A Narrative Review

**DOI:** 10.3390/bioengineering10101145

**Published:** 2023-09-29

**Authors:** Samuel E. Udeabor, Anja Heselich, Sarah Al-Maawi, Ali F. Alqahtani, Robert Sader, Shahram Ghanaati

**Affiliations:** 1Department of Oral and Maxillofacial Surgery, College of Dentistry, King Khalid University, Abha 62529, Saudi Arabia; seudeabor@kku.edu.sa; 2Department of Oral, Cranio-Maxillofacial, and Facial Plastic Surgery & Frankfurt Orofacial Regenerative Medicine (FORM) Lab, Johann Wolfgang Goethe University, 60590 Frankfurt am Main, Germany; anja.heselich@kgu.de (A.H.); sarah.al-maawi@kgu.de (S.A.-M.); robert.sader@kgu.de (R.S.); 3Department of Periodontics and Community Dentistry, King Khalid University, Abha 62529, Saudi Arabia; aaljabbar@kku.edu.sa

**Keywords:** socket healing, extraction socket, alveolar bone modeling, socket preservation, mini review, narrative review

## Abstract

The concept of extraction socket healing has been severally researched and reported over the years, since tooth extraction remains one of the most common procedures performed in the dental clinic. Understanding this healing process is of utmost importance because the outcome has a direct bearing on future prosthetic rehabilitation and, by extension, on patients’ esthetics and masticatory function, among others. This mini review, therefore, summarized the current knowledge on the different stages of socket healing, including the biologic and clinical events that occur following tooth extraction up until the complete closure of the socket. Additionally, the modeling of the alveolar bone/process post extraction, and the resultant dimensional changes that, altogether, shape the bone, were reviewed and documented. The effects of various socket preservation interventions to mitigate these dimensional changes, and therefore preserve the alveolar process in a condition suitable for future prosthetic rehabilitation, were highlighted. Finally, a review of some of the factors that influence the entire process was also carried out.

## 1. Introduction

Socket healing has been an important topic for study and research over the years because tooth extraction is one of the most common procedures carried out in the dental office [1,2]. It has been established that, following tooth extraction, the alveolar bone and, indeed, the surrounding soft tissue undergo a series of changes to fully restore the alveolar socket wound [1,2,3,4]. This entire event, that occurs after extraction and results in the complete healing and restoration of the socket, is referred to as socket healing [4].

This healing occurs in stages and the process begins immediately after tooth extraction up until around 6 months afterwards [5]. However, it has been established through various studies that the modeling and remodeling of alveolar bone continues for even longer than one year post extraction [4,6,7,8].

Socket healing is thought to be influenced by several factors, including local and systemic factors, iatrogenic, and even environmental factors [3,5]. These factors, together with differences in individuals in terms of inherent healing potentials, are believed to mold the outcome of socket wound healing [5]. Overall, the socket healing process results in changes to the alveolar bone in terms of loss of volume and various shape alterations [4,9,10,11]. In most cases, the alveolar ridge would be shorter and narrower with more buccal and labial resorption, which would negatively influence the outcome of future implant or other prosthetic rehabilitations [2,4,11].

In the international scientific literature, the topic of the healing of the alveolar socket after tooth extraction has been the subject of interest for implant purposes. For this reason, socket preservation surgery was born with the aim of conditioning and reducing bone contraction after tooth extraction [12,13,14,15]. Many of these authors and researchers are of the opinion that the immediate institution of socket preservation techniques with a range of biocompatible materials would go a long way in mitigating, or at least reducing, the dimensional changes that occur in the alveolar ridge following tooth extraction and making it suitable for implant restoration [4,7,16,17,18].

This, therefore, necessitates a full understanding of every aspect of socket healing, hence the present mini review intended to highlight the current knowledge on the subject in question.

## 2. Stages of Socket Healing

Four stages of socket healing are identifiable, and although there are some degrees of overlap in their timings, they can be clearly differentiated into the hemostasis and coagulation, inflammatory, proliferative, and remodeling stages (Figure 1) [6,19,20,21]. It has also been demonstrated in a previous study that the rate of healing varies between different individuals or subjects [6]. The four stages of healing actually progress at a relatively fast rate in humans with the formation of lamellar bone and marrow; however, the remodeling of this newly formed bone thereafter progresses at a comparatively slower rate and could last for even years after the tooth extraction [4,6,8].

## 3. Hemostasis and Coagulation

Following tooth extraction, the socket immediately fills with blood, leading to the formation of a blood clot, which is classically composed of red and white blood cells with platelets all enmeshed in a fibrin network [6,19,21]. In the first 7 days, this blood clot is replaced by granulation tissue which is mainly composed of a large number of blood vessels embedded in the connective tissue of mesenchymal cells and leukocytes [6,22]. Trombelli et al. (2008) [6], in their study on the modeling and remodeling of the human alveolar socket, noted that biopsies obtained from extraction sockets at 2–4 weeks post extraction had mainly mesenchymal cells with only a few red blood cells. This suggests that the initial clot that filled the extraction socket could have been completely remodeled within the first one week after extraction.

The process of clot formation in the socket follows the already established coagulation cascade. Bleeding following tooth extraction will lead to the interaction of the platelets with the exposed endothelial cells and extracellular matrix, leading to platelet aggregation and subsequent fibrin clot formation [20]. In addition to helping to achieve hemostasis, this initial blood clot that fills the socket also provides a framework or scaffold for the adhesion of the cells that will play important roles in the other stages of socket healing [20,23]. The blood clot and activated platelets, together with endothelial cells and leukocytes, release various cytokines and growth factors that modulate the inflammatory stage of socket healing [20,23,24].

## 4. Inflammatory Stage

During this stage of socket healing, which begins within 48 to 72 h after extraction, there is recruitment, migration, differentiation, and proliferation of inflammatory cells in response to the released cytokines and growth factors [4]. Large numbers of these inflammatory cells migrate to the healing socket and help to mop up debris, including the blood clot, in order to pave the way for new tissue formation [4,5].

Some of these cytokines and growth factors play different roles in wound healing. For example, platelet-derived growth factor (PDGF) and IL-1 are known to attract neutrophils to wound sites; transforming growth factor-beta (TGF-ß) is believed to help in converting circulating monocytes to macrophages and platelet-released vascular endothelial growth factors (VEGF) and macrophage-released fibroblast growth factor (FGF) also aid in extracellular matrix (ECM) formation and angiogenesis [24,25]. The roles of these growth factors and cytokines and, indeed, many others, as it relates to tooth socket healing has been examined and documented previously [6,26,27], however, according to Araujo et al. (2015) [4], a “simplistic characterization” of their effects is not appropriate due to their multiple and overlapping functions.

Neutrophils predominate in the early stages, followed by macrophages and later lymphocytes [20,23,24,25]. These cells phagocytose the blood clot and necrotic tissues [23]. The macrophages additionally release various growth factors, like the fibroblast growth factor (FGF), TGF-alpha, TGF-ß, and epidermal growth factor (EGF), which activate fibroblasts and osteoblasts as the socket healing progresses [25].

There is also, at this stage, the organization of the fibrin clot and replacement with granulation tissue which happens within the first 4 weeks after tooth extraction [4,6]. This granulation tissue is mainly composed of large numbers of new vessels with inflammatory cells and immature fibroblasts forming the connective tissue [4,5,6,19].

## 5. Proliferative Stage

Fibroplasia, which is the rapid deposition of the provisional matrix, marks the beginning of this proliferative stage and it is believed to be activated by TGF-ß1 and FGF-2 [20]. This stage of socket healing is often reported as occurring in two phases: the above-mentioned fibroplasia and woven bone formation, where the provisional matrix is invaded by newly formed blood vessels, bone forming cells, and a laying down of woven bone around the blood vessels [4,20]. The provisional matrix will progressively replace the granulation tissue and any remnants of the periodontal ligaments as this stage progresses [20]. This provisional matrix has been shown to be composed mainly of densely packed mesenchymal cells in a connective tissue matrix rich in collagen with numerous blood vessels and a few mononuclear leukocytes [6].

The presence of abundant blood vessels within the provisional matrix and osteoprogenitor cells results in woven bone being laid down especially around the vascular structures [4,20]. These finger-like projections of woven bone will eventually surround the blood vessels, giving rise to the primary osteon or the Harversian system [28,29]. The woven bone is basically mineralized tissue in a connective tissue matrix lined by osteoblasts and containing large numbers of osteocytes [6].

It is believed that almost all the granulation tissue is replaced by woven bone within 6 to 8 weeks of socket healing [5,6], but they are identified as early as 2 weeks post extraction [4]. In 27 human post extraction socket biopsies, Trombelli et al. (2008) [6] demonstrated that woven bone occupied a mean value of 34.0 ± 24.6% of the entire specimen analyzed. Bone morphogenic proteins (BMP), together with TGF-ß, have been shown to play major roles in this stage of bone morphogenesis and osteoblast differentiation [6,29].

It is important to note that sources of osteoblasts in extraction socket healing have been demonstrated in various studies to include periosteum, bone marrow, periodontal ligaments, adipocytes, and pericytes [6,30,31,32,33].

## 6. Modeling and Remodeling of Extraction Socket and Alveolar Bone

This represents the last stage of socket healing leading to the replacement of the woven bone with mature bone; in this case, lamellar bone and bone marrow which has load-bearing capacity [4,20]. Modeling generally refers to a change in bone structure with the actual modification of its shape and architecture, whereas remodeling is a change to the bone structure without an actual modification of its shape or architecture [4,6,20]. In this context, therefore, the laying down of lamellar bone and bone marrow to replace the woven bone within the healing socket is remodeling, while the dimensional changes that occur to the alveolar bone as a result of resorption is because of modeling [4,6,20]. The duration for bone remodeling varies in individuals and may take several months and years [4,6].

The laying down of mature mineralized trabeculae bone and marrow to replace the woven bone commences apically in the early phases of socket healing (around week 4) and then in the coronal area at about the 12th week [34] with the resultant sealing of the socket with cortical bone. Evian et al. [35], based on histological evidence, documented 8 to 12 weeks as the period when the new bone undergoes maturation and forms trabecular pattern. This is similar to the study by Ahn and Shin [36] who reported a complete socket healing with mineralized bone after 10 weeks post extraction. The above findings were further confirmed by various other studies that analyzed biopsies harvested from healing extraction sockets in week 12 post extraction and found evidence of mineralized trabecular bone in varying amounts [34,37,38].

However, Trombelli et al. [6] presented a slightly different picture with their study on biopsies from human alveolar sockets, reporting that woven bone was the dominating tissue (41%) within 12 to 24 weeks post extraction and that lamellar bone and marrow were found only in 1 out of their 11 biopsies in the same period. A higher percentage, of up to 65% by volume of lamellar bone and bone marrow to the total tissue, was reported by Lindhe et al. (2012) [39] after 16 weeks of extraction socket healing. Table 1 details the studies analyzed with the reported healing duration.

The activities of osteoblasts and osteoclasts are responsible for the modeling and remodeling of the alveolar socket [4,6] and a host of growth factors, like runt-related transcription factor 2 (RUNX2), osteocalcin (OC), osteopontin (OPN), osterix (OST), receptor activator of nuclear kappa B (RANK), receptor activator of nuclear kappa B ligand (RANKL), and osteoprotegerin (OPG), are also believed to be essential in modulating this phase [44].

The outcome of modeling and remodeling will, therefore, result in dimensional changes in the edentulous alveolar ridge with an overall reduction in ridge height and volume [2,4,47,48,49].

## 7. Dimensional Changes in Alveolar Bone following Socket Healing

The modeling of the alveolar bone post extraction occurs on every aspect of the alveolar bone: buccal, labial, lingual, and palatal, leading to changes in the dimensions of the bone. The bundle bone is reported to be the first to resorb [50], and there is an overall rapid reduction in the alveolar ridge size in the first 6 months after extraction, thereafter, resorption continues at a slower pace throughout life [50,51,52]. This fast resorption is thought to be due to the high osteoclastic activity that is present during the early phases of the modeling and remodeling of the alveolar bone post extraction [20]. Schropp et al. (2003) [8] also demonstrated this in a 12-month prospective study that clinically and radiographically examined hard and soft tissue changes after tooth extraction. They reported a bucco-lingual and bucco-palatal width reduction of about 50% in the first 3 months of socket healing (Figure 2). This resorption, according to the authors, was more pronounced in the molar region when compared to the premolar region and in the mandible more than the maxilla.

The labial and buccal surfaces of the alveolar bone exhibit more resorption in comparison to the lingual/palatal surfaces [2,4,41]. In fact, a more vertical bone loss in the buccal surface was demonstrated by Araújo et al. [4,41] due to the lingual bone being generally wider than the buccal bone. Similarly, modeling resulted in a more horizontal bone resorption, especially in the labial aspect of the alveolar bone [53,54,55,56].

The effect of modeling on the alveolar ridge would, therefore, leave the ridge shorter and narrower with a more buccal/labial resorption and a resultant more lingually/palatally positioned alveolus (Figure 2) [2,36,57]. In a systematic review that included 12 publications, Van der Weijden et al. [2] documented, on average, a clinical loss in width of 3.87 mm and a loss of height of 1.67–2.03 mm clinically/1.53 mm radiographically following tooth extraction socket healing. Tan et al. [52], in another systematic review, showed that, at 6 months post extraction, there was a horizontal bone loss of 29–63% and a vertical loss of 11–22%.

**Figure 2 bioengineering-10-01145-f002:**
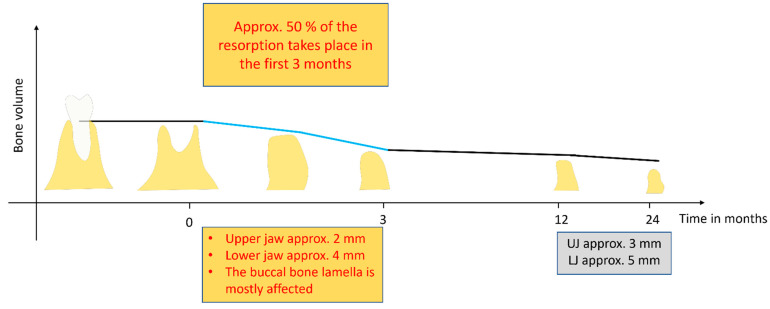
Overview of alveolar bone loss in the first 1 to 2 years after tooth extraction based on the literature [8,52,58]. (Approx.: approximately; UJ: upper jaw; LJ: lower jaw).

## 8. Factors That Affect Socket Healing

The studies reviewed show a great deal of variance in terms of periods of bone tissue formation and maturation, and the outcome of modeling and remodeling [6,19,22,40]. This variation in the healing of the alveolar socket and the eventual outcome may be influenced by a host of factors that range from local, systemic, iatrogenic, and even environmental factors [3,5,6,40].

The local factors reported to influence socket healing include the site of extraction [40]. Molar tooth extraction sites, especially mandibular molars, show the highest erratic healing in comparison to maxillary incisor/canine sites that show the least [40]. This was further corroborated in a retrospective review by Pramstraller et al. [59], where molar sites showed more bone resorptions than premolar sites. They claimed that this may be due to the fact that posterior teeth generally present more difficulty with extraction and also leave a wider socket afterwards.

Another local factor thought to influence alveolar socket healing is the raising of the full muco-periosteal flap before extraction. This has been shown to cause the loss of attachment and compromise blood supply to the healing socket and may, therefore, lead to bone resorption [5,60]. Additionally, multiple edentulous sites are also reported to show more resorption of the alveolar bone than single edentulous sites [59].

The molecular and cellular events that result in socket healing post extraction can also be affected by some systemic factors like smoking [61,62,63], uncontrolled systemic diseases like diabetes mellitus [46,64], post-menopausal osteoporosis [52], and other systemic factors. The precise mechanism through which smoking alters socket healing is not well understood, but Saldanha et al. [61] observed a 0.5 mm higher alveolar crest reduction in smokers versus non-smokers. They believed that nicotine, which is a major component of tobacco smoke, may be partly responsible because it is a cytotoxic and vasoactive substance. It has also been suggested in the literature that post-menopausal osteoporosis may have a causal effect on residual ridge reduction post extraction, with a resultant smaller maxillary alveolar ridge and a knife-edge mandibular ridge [52].

Similarly, Delvin et al. [64] observed the inhibition of collagen framework formation in the healing socket of uncontrolled insulin-dependent diabetes, leading to more alveolar bone resorption. This poor socket healing peculiar to patients with uncontrolled type 2 diabetes may improve with hyaluronic acid treatment of the socket post extraction [65]. Hyaluronic acid has been shown to improve early phases of extraction socket healing by encouraging cell migration, proliferation, and differentiation [66]. Similarly, the use of chlorhexidine mouth rinses after tooth extraction has been shown to limit alveolar bone resorption, although the exact mechanism is not clearly understood [5].

In addition to its effect on the management of jaw osteoradionecrosis, hyperbaric oxygen therapy has been shown by Liao et al. (2020) [67] to facilitate the healing of the extraction socket and promote alveolar ridge preservation. The authors were able to demonstrate through animal experimental study that hyperbaric oxygen therapy reduced alveolar bone resorption post extraction by promoting the formation of osteoblasts and reducing osteoclast formation among other functions. They are, therefore, of the opinion that this may be extrapolated to human patients in the clinic to hasten alveolar socket healing and also aid alveolar ridge preservation.

It is also important to consider the degree of damage incurred to the tissue during extraction by the surgeon and the resultant bony defect as factors that could influence the outcome of alveolar socket healing, as such cases would result in more bone resorption than less traumatic cases [2,5,6].

## 9. Effects of Alveolar Ridge Preservation Techniques on Socket Healing Outcome

The idea of alveolar ridge preservation (ARP) or socket preservation was borne out of the need to counter the volumetric loss of alveolar bone subsequent to tooth extraction with the intent to present a ridge that is adequate both in width and height for implant and other prosthetic rehabilitations purposes [12,13,14]. It is established that the effects of resorption and the remodeling of alveolar bone after tooth extraction could result in a loss of up to half of the original volume of the bone in as little as 12 weeks (Figure 2) [8,12,14] and so any intervention procedure that will slow or mitigate these effects would go a long way in improving prosthetic outcomes. The proponents of alveolar ridge preservation (ARP) techniques are of the opinion that the placement of a graft material in fresh extraction socket would help in stabilizing the blood clot in the early phases of socket healing, act as a scaffold throughout the socket healing period to encourage the osteoconduction of the newly formed bone, and be gradually resorbed as newly formed bone is laid down [5,14,20].

Over the years, myriad graft materials have been used for alveolar ridge preservation procedures, both in in vivo animal experiments and human studies, and these biomaterials include autogenous grafts, allografts, xenografts, and grafts based on alloplastic materials [12,14,20,68,69,70]. Various clinical studies employing different graft materials for alveolar ridge preservation have yielded positive results. Most of these studies agree that ARP is efficient in diminishing the bone resorption that occurs post extraction and hence prevent the adverse dimensional changes that would ordinarily negate future implant treatment [45,71,72].

Alveolar ridge preservation could be achieved via socket grafting, in which case different types of bone substitute materials or even autologous blood-derived products like PRF are used to fill the extraction socket immediately after extraction [14]. ARP could also be achieved through a socket sealing procedure where the socket is covered by a barrier material to prevent soft tissue ingrowth into the socket and encourage bone regeneration [73]. Additionally, a combination of socket grafting and socket sealing with a barrier material is another method of ARP [73]. These techniques, especially socket sealing and the combination procedure, according to MacBeth et al. (2022) [16] in a randomized clinical trial, limited vertical bone loss four months post extraction when compared to non-treated sockets. This was the uniform report by different authors who found that ARP techniques, in comparison to non-intervention, limited contour changes [74], diminished the physiologic resorption process [71], reduced both the horizontal and vertical shrinkage [68], and overall minimized the dimensional changes that occur after tooth extraction as a result of bone resorption [14].

Despite these findings and their supposed efficacy in alveolar ridge preservation, Araújo et al. (2015) [4], in a review article, summarized that the immediate placement of the implant, or, indeed, different graft materials in fresh extraction sockets for the purpose of ARP, do not prevent buccal bone resorption. They opined that the socket grafts instead act only to compensate for this bone loss and to encourage new bone formation as well.

## 10. Conclusions

The healing of extraction socket involves four overlapping stages/phases that are modulated by various growth factors and cytokines. A complete sealing of the socket with matured mineralized trabecular bone and marrow is achieved around the 12th week on average. However, the resorption of the alveolar process and bone remodeling continue throughout life. This entire healing process is nevertheless influenced by several local, systemic, and iatrogenic factors, leading to varied unfavorable outcomes in different patients.

The dimensional changes that occur subsequent to tooth extraction result in more vertical and horizontal bone loss in the buccal and labial surfaces when compared to the buccal and palatal surfaces. This loss in the shape and overall volume of alveolar bone would negatively impact the fabrication of different prostheses, thereby influencing esthetics and function, hence the introduction of different alveolar ridge preservation interventions to ameliorate these negative outcomes of socket healing. Currently, different ARP techniques are in clinical use and are constantly evolving. Many of these have proven efficient in compensating for the dimensional changes due to post extraction resorption and presenting an alveolus with adequate volume and contour for implant restoration and other prosthetic rehabilitation purposes.

## Figures and Tables

**Figure 1 bioengineering-10-01145-f001:**
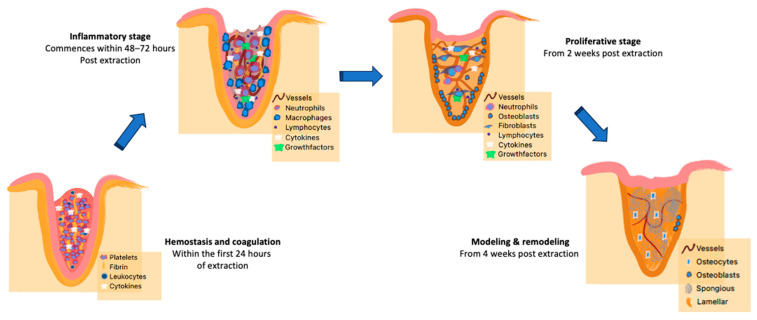
Graphical representation of the 4 stages/phases of socket healing from the time of tooth extraction, detailing the timing of the process and the physiological events involved.

**Table 1 bioengineering-10-01145-t001:** Characteristics of studies that evaluated alveolar socket healing in both humans and animals.

S/No.	Study	Year	Study Design	Model	Tooth Type	Healing Period/Outcome	Evaluation Method
1	Trombelli et al. [6]	2008	Clinical study	Human	Single-rooted teeth	12–24 weeks (Dominant tissue: Provisional matrix and woven bone)	Immunohistochemistry (IHC)
2	Delvin and Sloan [19]	2002	Clinical study	Human	Premolars	Not reported (Final tissue analysis was 2 weeks post extraction)	IHC
3	Kim et al. [40]	2014	Retrospective Explorative study	Human	All teeth	5.5 ± 2.5 months	Panoramic radiography and CT scan
4	Srinivas et al. [22]	2018	Clinical study	Human	All teeth	3 months	Cone beam computed tomography (CBCT)
5	Nahles et al. [34]	2013	Clinical study	Human	All teeth	12 weeks (complete ossification in some patients)	IHC
6	Ahn and Shin [36]	2008	Clinical study	Human	All teeth	10 weeks (complete socket healing)	Histology
7	Heberer et al. [38]	2011	Clinical study	Human	All teeth	12 weeks (44% bone formation)	Histology
8	Lindhe et al. [39]	2012	Clinical study	Human	Posterior maxillary teeth	Not specified (varied)	Histology
9	Cardaropoli et al. [7]	2005	Experimental study	Dog	Premolar	3 months (complete healing)	Histology
10	Araújo and Lindhe [41]	2005	Experimental study	Dog	Premolars	4–8 weeks (Dominant tissue: Mineralized bone and bone marrow)	Histology
11	Vignoletti et al. [42]	2012	Experimental study	Dog	Premolars	6 weeks (Socket closed with cortical bone)	Histology
12	Sheng et al. [43]	2023	Experimental study	Rat	Molars	4 weeks	Immunofluorescence, Micro-CT, IHC, RT-PCR
13	Hassumi et al. [44]	2018	Experimental study	Rat	Incisors	28 days	Micro-CT, IHC, RT-PCR
14	Yugoshi et al. [45]	2002	Experimental study	Rat	Incisors	3 weeks (Large volumes of trabecular bone formed)	Histology
15	Younis et al. [46]	2013	Experimental study	Rabbit	Incisors	4 weeks	IHC
16	Scala et al. [28]	2014	Experimental study	Monkey	Premolars	90–180 days (Dominant tissue: Trabecular bone and bone marrow)	Histology

## Data Availability

Data sharing not applicable to this article.

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
