# Peer review of "Current Knowledge on the Healing of the Extraction Socket: A Narrative Review"

_bioengineering, 2023, doi:10.3390/bioengineering10101145_

Round 1

Reviewer 1 Report

Healing of extraction sockets is a well-studied topic in dentistry and there is a significant body of literature available on this subject. Yet the studies in the paper discuss the primary research articles on healing without intervention. Hence there are several studies available that focus on the intervention and I suggest to integrate some key articles as this is routinely done in everyday practice:

1.      Avila-Ortiz, G., Chambrone, L., & Vignoletti, F. (2014). Effect of alveolar ridge preservation interventions following tooth extraction: a systematic review and meta-analysis. Journal of Clinical Periodontology, 41(8), 820-830. https://pubmed.ncbi.nlm.nih.gov/30623987/

This systematic review and meta-analysis evaluated different interventions used for alveolar ridge preservation following tooth extraction. It provides an overview of the effectiveness of interventions in preserving the dimensions of the extraction sockets.  

2.       Araújo, M. G., Linder, E., & Lindhe, J. (2009). Bio-Oss collagen in the buccal gap at immediate implants: a 6-month study in the dog. Clinical Oral Implants Research, 20(7), 689-695. https://pubmed.ncbi.nlm.nih.gov/21091538/

This animal study investigated the effectiveness of Bio-Oss collagen for socket preservation and implant placement. It evaluated the bone healing outcomes of immediate implants placed in extraction sockets augmented with Bio-Oss collagen.  

3.       Fickl, S., Zuhr, O., Wachtel, H., Stappert, C. F., & Stein, J. M. (2008). Dimensional changes of the alveolar ridge contour after different socket preservation techniques. Journal of Clinical Periodontology, 35(10), 906-913. https://pubmed.ncbi.nlm.nih.gov/18713258/

This study compared the dimensional changes of the alveolar ridge contour after different socket preservation techniques. It assessed the effectiveness of various socket preservation interventions in maintaining the ridge dimensions following tooth extraction.  

These articles provide a range of evidence on various intervention methods used for healing extraction sockets. It is always advisable to review the full papers for a comprehensive understanding of the interventions, methodologies, and outcomes discussed in these studies. You can explore a few literature sources to understand the effect of hyaluronic acid and antibiotics on the healing of extraction sockets:  

Talking about current knowledge and influencing factors these topics need to be discussed in my opinion.

Author Response

Thank you for your meticulous review of our manuscript and the valuable suggestions. 

We have added a section on intervention as you suggested, where we summarized alveolar ridge preservation (pages 7 and 8) based on your recommended articles and other relevant literature.

We have also added the effect of hyaluronic acid on healing of extraction sockets in the section of "factors that affect socket healing" (page 7).

Overall, we have increased the length of the review.

All the corrections and additions are highlighted.

Reviewer 2 Report

Dear authors

the manuscript is well written and organized. Beside brief, gives the reader a nice sum of the post-extractive events.

well done

Author Response

Thank you for your review and kind words. We appreciate.

We have increased the length of the manuscript as a whole and added a segment on alveolar ridge preservation as it affects socket healing.

Reviewer 3 Report

Many thanks for the paper submission. This is an interesting paper that deals with the current knowledge of healing of the extraction socket. Today most of the research in this field is related to the manipulation of the socket with biomaterials so, with the purpose of the journal, I support the idea that this concept should be emphatized in the introduction section. There are some major flaws in the article that should be corrected.

1) keywords: please place at least 6 keywords and put socket preservation among them

2) the abstract should emphatize more the novelty of this article

3) Introduction is very short: please increase in leght. After this phrase "...These factors together with differences in individuals in terms of inherent healing potentials are believed to mold the outcome of socket wound healing.." please add this short paragraph in order to increase the scientific soundness and clinical interest for the readers:

"...In international scientific literature, the topic of healing of the alveolar socket after tooth extraction has been the subject of interest for implant purposes. For this reason, the socket preservation surgery was born with the aim of conditioning and reducing bone contraction after tooth extraction.."

please cite the following:

MacBeth ND, Donos N, Mardas N. Alveolar ridge preservation with guided bone regeneration or socket seal technique. A randomised, single-blind controlled clinical trial. Clin Oral Implants Res. 2022 Jul;33(7):681-699. doi: 10.1111/clr.13933. Epub 2022 Jun 22. PMID: 35488477; PMCID: PMC9541021.   MacBeth N, Trullenque-Eriksson A, Donos N, Mardas N. Hard and soft tissue changes following alveolar ridge preservation: a systematic review. Clin Oral Implants Res. 2017 Aug;28(8):982-1004. doi: 10.1111/clr.12911. Epub 2016 Jul 26. PMID: 27458031.   Chisci G, Hatia A, Chisci E, Chisci D, Gennaro P, Gabriele G. Socket Preservation after Tooth Extraction: Particulate Autologous Bone vs. Deproteinized Bovine Bone. Bioengineering (Basel). 2023 Mar 27;10(4):421. doi: 10.3390/bioengineering10040421. PMID: 37106608; PMCID: PMC10136074.

Chisci G, Fredianelli L. Therapeutic Efficacy of Bromelain in Alveolar Ridge Preservation. Antibiotics (Basel). 2022 Nov 3;11(11):1542. doi: 10.3390/antibiotics11111542. PMID: 36358197; PMCID: PMC9687015.   4) Figure 1: this image should be formatted on the basis of MDPI.

Author Response

We truly appreciate your meticulous review of our manuscript and your valuable suggestions.

We have emphasized the place of socket preservation in the abstract and introduction as you suggested.

  1. The keywords have been increased to 6 and socket preservation included among them.
  2. The abstract has been extended and the importance of socket preservation in extraction socket healing emphasized.
  3. The length of the introduction has been increased and the suggested phrase added. We have added the recommended articles and indeed other relevant literature.
  4. However, we did not understand how to format the images on the basis of MDPI.
  5. We have increased the overall length of the manuscript and added a segment on alveolar ridge preservation as it affects socket healing outcome.
  6. The corrections and additions are highlighted in blue.

Round 2

Reviewer 3 Report

Accept